# Ascorbic Acid and Graphene Oxide Exposure in the Model Organism *Acheta domesticus* Can Change the Reproduction Potential

**DOI:** 10.3390/molecules29194594

**Published:** 2024-09-27

**Authors:** Barbara Flasz, Monika Tarnawska, Andrzej Kędziorski, Łukasz Napora-Rutkowski, Joanna Szczygieł, Łukasz Gajda, Natalia Nowak, Maria Augustyniak

**Affiliations:** 1Institute of Biology, Biotechnology and Environmental Protection, University of Silesia in Katowice, 40-032 Katowice, Poland; barbara.flasz@us.edu.pl (B.F.); monika.tarnawska@us.edu.pl (M.T.); andrzej.kedziorski@us.edu.pl (A.K.); lgajda@us.edu.pl (Ł.G.); nowaknatalia6499@gmail.com (N.N.); 2Polish Academy of Sciences, Institute of Ichthyobiology and Aquaculture in Gołysz, 43-520 Zaborze, Poland; lukasz.napora-rutkowski@golysz.pan.pl (Ł.N.-R.); joanna.szczygiel@golysz.pan.pl (J.S.)

**Keywords:** ascorbic acid, graphene oxide, *Acheta domesticus*, vitellogenin

## Abstract

The use of nanoparticles in the industry carries the risk of their release into the environment. Based on the presumption that the primary graphene oxide (GO) toxicity mechanism is reactive oxygen species production in the cell, the question arises as to whether well-known antioxidants can protect the cell or significantly reduce the effects of GO. This study focused on the possible remedial effect of vitamin C in *Acheta domesticus* intoxicated with GO for whole lives. The reproduction potential was measured at the level of Vitellogenin (*Vg*) gene expression, Vg protein expression, hatching success, and share of nutrition in the developing egg. There was no simple relationship between the *Vg* gene’s expression and the Vg protein content. Despite fewer eggs laid in the vitamin C groups, hatching success was high, and egg composition did not differ significantly. The exceptions were GO20 and GO20 + Vit. C groups, with a shift in the lipid content in the egg. Most likely, ascorbic acid impacts the level of *Vg* gene expression but does not affect the production of Vg protein or the quality of eggs laid. Low GO concentration in food did not cause adverse effects, but the relationship between GO toxicity and its concentration should be investigated more thoroughly.

## 1. Introduction

Due to auspicious physiochemical, mechanical, and electrical properties, in the 1950s, a new industry was born—nanotechnology—the industry of engineered nanomaterials. Graphene oxide (GO) is one of the most attractive graphene derivatives, widely used in many fields of human life. Graphene-based nanomaterials are commonly used in many applications, including sensors, biosensors, electronic devices, supercapacitors, spintronics, photonics, flexible and next-generation electronics, biomedical applications, energy storage, and solar cells [1,2,3,4,5]. Graphene nanomaterials are also used in medicine and biology applications [6,7] and in agriculture [8,9]. The growing global nanomaterials market brings to light many important questions about its possible toxicity to humans, animals, and other organisms that build ecosystems. The use of nanoparticles in the industry carries the risk of their release into the environment and water, soil, and air pollution. Thus, a thorough understanding of the biological consequences of nanomaterials like GO is imperative.

One proposed GO toxicity mechanism is reactive oxygen species (ROS) production, which can lead to an imbalance between free radicals and antioxidants [10]. In intracellular signaling paths, free radicals can be secondary messengers that can cause severe disorders, for instance, protein denaturation, DNA fragmentation, mitochondrial dysfunctions, and lipid bilayer degradation [10,11]. The increase in dose and GO exposure may affect a decrease in antioxidative enzymes, like SOD and GSH, that are essential for ROS elimination [10]. High free radicals can activate signaling pathways that result in apoptosis and necrosis [12].

Based on the presumption that the primary GO toxicity mechanism is ROS production in the cell, the question arises as to whether well-known antioxidants can protect the cell or significantly reduce the effects of GO. Many chemicals can be good candidates for ROS elimination because the high reactivity of free radicals results in extracting an electron from almost any available molecule. However, reacting to reactive oxygen species is insufficient to be an effective antioxidant. The efficient antioxidant (a) must be present in an adequate amount in the cell, (b) it must react with a variety of free radicals, and (c) it must be suitable for regeneration [13,14]. One of the potent antioxidants is vitamin C (ascorbic acid, Vit. C), the most prevalent antioxidant and famous for scavenging free radicals. It is known for promoting growth in life expectancy, protecting from the toxic effects of many chemicals, and significantly improving the potential of reproduction [15,16]. There are studies on the potential protecting role of vitamin C from the harmful effects of nanoparticle action. In the Wistar rats, the role of ascorbic acid in reducing oxidative stress caused by zinc oxide nanoparticles was investigated. The results showed a dose-dependent reduction in the toxic effects caused by zinc oxide nanoparticles in liver enzymes and blood, indicating that oxidative stress is one of the nanoparticle mechanisms [17]. Another study on rats showed that acute pulmonary oxidative stress induced by zinc oxide nanoparticles was suppressed by supplying ascorbic acid [18]. Exposition to water with multi-walled carbon nanotubes in red spinach overnight and then with vitamin C medium presented plants protected from the oxidative stress caused by the nanoparticles. Here, the authors suggested that nanoparticles trigger oxidative stress in the organism [19]. In the literature, there are examples of the antioxidant capacity of ascorbic acid in rationalizing reproductive potential. In cadmium-treated male rats, vitamin C helped prevent the decrease in the 3β-hydroxysteroid dehydrogenase enzyme (HSD), which suggests the protective role of ascorbic acid against testicular damage [20]. Ascorbic acid supplementation reduced the free radical level in rabbit semen and improved its quality [16]. Even though vitamin C is a potent antioxidant and reduces free radicals, little research has been conducted to assess whether it can effectively reduce oxidative stress caused by nanoparticles. On the other hand, it should also be noted that vitamin C does not always bring protective or remedial effects. In our previous project [21], it was presented that house crickets treated with ascorbic acid for 15 days after long GO intoxication had no evident remedial effect. In some studies, synergistic toxicity of ascorbic acid and other molecules was shown [22], which is a reason to investigate the effects of vitamin C, especially in relation to graphene oxide.

The presented study focused on the possible remedial effect of vitamin C in invertebrate *Acheta domesticus* intoxicated with graphene oxide nanoparticles for their whole lives. This project concerned the impact on reproduction potential measured at the level of Vitellogenin (*Vg*) gene expression, Vg protein expression, hatching success, and share of nutrition in the developing egg. We decided to measure Vitellogenin, which is strongly connected with reproduction and is widely used as a biomarker of reproduction disorders [23,24,25]. All oviparous species accumulate yolk protein in the eggs during oogenesis. Vg expression studies are good candidates for measuring the possible protective or remedial effects of vitamin C supplemented with GO-treated house crickets.

## 2. Results

### 2.1. Vg Gene Relative Expression

Vitellogenin relative expression in fat body tissue was assessed on the 5th and 15th days of adulthood. On the 5th day, significant differences in *Vg* gene expression were observed between groups. The group treated with Vit. C exhibited the highest relative *Vg* gene expression compared to other groups. This level was significantly different from the control, GO20, and GO20 + Vit. C groups. Both GO20 and GO20 + Vit. C groups showed lower *Vg* gene expression levels compared to the control, although these differences were not statistically significant. Surprisingly, the group fed with a combination of GO20 + Vit.C displayed the lowest *Vg* mRNA expression level (Figure 1).

By the 15th day, the differences between groups were less pronounced, with no significant differences observed (Figure 1).

### 2.2. Total Vg Protein Expression

The total Vg protein content in the fat body of house crickets was measured on the 5th and 15th days of adulthood (Figure 2). Vg expression is known to be specific to factors such as sex, tissue, hormones, and stage of development [26,27]. Control groups on the 5th and 15th days of adulthood showed differences in Vg protein expression, with higher levels detected on the 5th day, although not statistically significant.

On the 5th day, Vg protein expression among the investigated groups was similar. The highest expression was observed in the GO20 + Vit. C group, although not statistically significant. By the 15th day, Vg protein content in the Vit. C, GO20 + Vit. C, and GO20 groups was significantly higher than in the control group. No differences were observed between corresponding groups when comparing the 5th and 15th days.

### 2.3. Vg Protein Precursor and Subunits Expression

In the analysis of A. domesticus Vg protein, including its precursor and subunits [28], the control group from the 5th day of adulthood served as the reference for comparison with treated groups (Figure 3A). On this day, the control group showed higher expression of the Vg protein precursor compared to the treated groups, with expressions in the Vit. C, Vit. C + GO20, and GO20 groups ranging from 14% to 18% lower than the control. However, the Vg 130 subunit exhibited different expression patterns (Figure 3C), with the control group showing lower expression compared to the treated groups. The highest expression of the 130 subunit was observed in the Vit. C group, followed by GO20 and GO + Vit. C groups. All treated groups displayed slightly higher expression of the 97 kDa subunit compared to the control (Figure 3E).

On the 15th day of adulthood, the precursor expression was higher in treated groups compared to the control (Figure 3B). The GO20 + Vit. C group exhibited the highest expression of the precursor. The 130 kDa subunit showed varying expression profiles among the investigated groups, with the control group displaying the lowest expression and GO20 showing the highest (Figure 3D). The 97 kDa subunit also presented varied results, with the control group exhibiting the lowest expression and the GO20 + Vit. C group showing the highest (Figure 3F).

### 2.4. Egg Laying and Hatching Success

Hatching success was assessed based on the number of eggs laid by the female, the number of larvae hatched, and the total number of hatched eggs per group (Figure 4). Both the parameters of eggs per female and hatching per female showed similar trends (Figure 4A). Control and GO20 groups exhibited the highest success in terms of laid eggs and hatched larvae, while GO20 + Vit. C and Vit. C groups displayed lower reproductive success. Notably, the GO20 + Vit. C group showed the lowest success in both parameters, with statistically significant differences compared to the control and GO20 groups. These results suggest that the treatments with GO, Vit. C, or both may have affected the egg-laying process rather than the hatching process.

Total hatching success, calculated as the number of hatched larvae divided by the total number of eggs, was higher in the treated groups compared to the control (Figure 4B). Specifically, groups treated with Vit. C, GO20 + Vit. C, and GO20 showed similar levels of hatching percent, with no significant differences observed among them. Conversely, the control group exhibited the lowest hatching success, with statistically significant differences observed compared to the treated groups.

### 2.5. Major Energetic Components in Eggs

Figure 5 illustrates the percentage distribution of major energetic components in insect eggs. Protein was found to be the most abundant component, constituting more than 95% of the egg composition across all examined groups. Glucose was present in the smallest proportion, accounting for no more than 0.06% of the total share. The contribution of glycogen varied between 0.5% and 1% of the total content. Lipids exhibited the greatest variability, ranging from 0.08% to 0.74% of the total composition. Notably, despite the increase in the percentage share of lipids in the experimental groups, it did not come at the expense of other energy and building components of the eggs. While the percentage composition of egg components changed, the actual content of these components (in mg∙mL^−1^) remained relatively consistent, except for lipids (Figure 5 and Figure 6).

#### 2.5.1. Lipids

On the 5th day of adult life, the Vit. C group showed slightly lower lipid content in eggs compared to the control, while the GO20 and GO20 + Vit. C groups had significantly higher lipid levels (Figure 6A). Similar trends were observed on the 15th day, with the GO20 + Vit. C group exhibiting the highest lipid content. Overall, lipid levels were significantly elevated in the GO20 and GO20 + Vit. C groups compared to the control, indicating the potential effects of GO and Vit. C on lipid accumulation in cricket eggs.

#### 2.5.2. Glucose

On the 5th day of adulthood, glucose levels were comparable between the control and GO20 + Vit. C groups (Figure 6B). Both the Vit. C and GO20 groups had similar glucose content, lower than the control and Vit. C groups. Despite these differences, there were no significant statistical variations between the groups. By the 15th day, all groups showed higher glucose levels than on the 5th day, but without statistical significance. Glucose content remained consistent across all investigated groups.

#### 2.5.3. Glycogen

On the 5th day of adulthood, glycogen levels were highest in the GO20 + Vit. C group, followed by the Vit. C group, the control group, and lowest in the GO20 group (Figure 6C). However, no significant differences were observed between the groups. By the 15th day, glycogen levels increased in all groups, with no significant differences observed. Notably, there were significant differences in glycogen levels between the control groups on both the 5th and 15th days, as well as within the GO20 group across the two time points.

#### 2.5.4. Total Protein

On the 5th day of adulthood, total protein levels in the control, GO20 + Vit. C, and GO20 groups were comparable, with no significant differences observed (Figure 6C). The Vit. C group exhibited slightly higher total protein content, though not statistically significant. By the 15th day, the control and Vit. C groups had similar total protein levels, while the GO20 group had lower protein content, and the GO20 + Vit. C group had the lowest. Despite some differences, no statistically significant variations were observed between the groups, both within each time point and when comparing the 5th and 15th days.

## 3. Discussion

In insects, Vg is expressed in fat body tissue, secreted into the hemolymph, and taken up by developing oocytes. It is stored as Vitellin (Vn) and serves as a vital nutrient for embryos, making vitellogenesis a potential biomarker for reproductive disruptions [25,26].

The therapeutic effect of ascorbic acid on insects exposed to long-term graphene oxide (GO) exposure was investigated, along with the protective effects of dietary vitamin C consumption. Previous research has shown that increasing concentrations of vitamin C in food are associated with higher egg and larvae production per female and better blood parameters [29]. A study on rainbow trout fed with vitamin C-free food revealed a shift in lipid metabolism and reduced levels of Vg, underscoring the importance of ascorbic acid consumption [30]. This study examined changes at different biological levels, including gene expression, protein activity, egg laying, hatching success, and egg composition. However, no simple relationship was found between *Vg* gene expression and Vg protein content measured using ELISA and Western Blot techniques.

In our study, we observed significantly higher relative *Vg* mRNA expression in the group treated with ascorbic acid compared to the control on the 5th day of adult life (Figure 1). However, there was no such correlation in Vg protein precursor expression or total Vg protein expression (Figure 2), suggesting that the mechanism of vitamin C action may involve the control of gene expression, including Vitellogenin.

Vitamin C is known for its antioxidant activity and role as a cofactor in biological processes and immune responses [15,31,32]. The other vital role of ascorbic acid is the modulation of gene expression [33]. One of the most studied regulatory roles of ascorbic acid is in collagen synthesis. It is well documented that in the production of collagen type I, ascorbic acid increases the level of procollagen mRNA in two ways: by stimulating the transcription of the procollagen gene and by limiting the degradation of procollagen mRNA [34,35,36]. Ascorbic acid also stimulates the synthesis of other genes, such as the acetylcholine receptor and cytochrome P-450 [37,38,39,40,41]. The described examples demonstrated that ascorbic acid may have a role at the transcriptional level. In our study, *Vg* mRNA expression in insects was high on the 5th day of adult life. On the 15th day, the *Vg* gene was expressed higher than the control but without statistical significance. This lower expression of *Vg* mRNA on the 15th day may be related to the nature of Vg, which is sex, tissue, and stage-specific [26,27,42]. As insects age, the intensity of *Vg* gene expression may decrease. Thus, the difference between the control and Vit. C group is not as straightforward as on the 5th day when insects were preparing for rapid reproduction. In the groups treated with GO20 or GO20 + Vit. C, we observed a downward trend in the expression of the *Vg* gene on the 5th day of adult life (Figure 1). This phenomenon raises the question: why is *Vg* downregulated?

The toxic effects of graphene oxide (GO) are well documented in the literature, primarily attributed to increased oxidative stress [43,44,45,46]. Reactive oxygen species (ROS), produced as a result of this stress, can act as secondary messengers, triggering specific cellular responses, including cytotoxicity [47,48,49]. GO toxicity can impact various organs, causing molecular, tissue, and organismal changes such as reduced survival rates, higher mortality, tissue damage, abnormalities, DNA damage, and altered gene expression [11,43,44,50,51,52,53,54,55]. In house crickets, Vg is expressed in fat body tissue, analogous to the liver in vertebrates. GO has been shown to accumulate in the liver of mice, causing ultrastructural changes like vacuolization and mitochondrial disruptions in hepatocytes and reducing DNA damage levels [55,56,57,58]. These studies suggest that while established doses of graphene might not be acutely toxic, it is crucial to determine safe exposure levels.

In our project, we used low concentrations of GO administered throughout the insects’ lifespans. This continuous exposure led to lower *Vg* mRNA expression in the fat body tissue. Based on the evidence that GO can accumulate in the liver, we hypothesize that long-term GO treatment in crickets allows accumulation in the fat body tissue directly or indirectly (via ROS as a secondary messenger), affecting *Vg* gene expression.

The relative expression of *Vg* in groups treated with GO20 + Vit. C was the lowest, indicating no therapeutic or protective effect of ascorbic acid on insects treated with GO (Figure 1). While human blood pH is tightly regulated and unaffected by vitamin C intake, vitamin C can acidify urine. Insects have an open circulatory system with hemolymph [59,60], which might not have stringent buffering systems like human blood. Consequently, consuming vitamin C could acidify the hemolymph. This acidification may alter the protein corona coating GO [61,62], exposing functional groups and potentially increasing GO’s reactivity and toxicity, leading to low *Vg* mRNA expression.

Despite the differences in *Vg* mRNA production on the 5th day, Vg protein levels were similar to controls for total Vg protein, precursor, and subunit expression (Figure 2 and Figure 3). However, by the 15th day, total Vg protein expression was significantly higher in all groups compared to controls. The Vg precursor showed a slight increase, and the subunits, especially the Vg 130 subunit, were expressed at least 100% higher (Figure 3). These pronounced differences on the 15th day likely result from protein accumulation in the fat body of mature *A. domesticus*. On the 5th day, the differences were less noticeable as Vg production had only recently begun.

The lack of correlation between protein abundance and mRNA expression is unsurprising [63,64,65].

Vitellogenin (Vg) protein produced in the fat body is transported in the hemolymph to the oocytes in the ovaries and stored as Vitellin (Vn) [26]. In our study, the average number of eggs laid and larvae hatched by females showed that the Vit. C and GO20 + Vit. C groups had the lowest egg-laying success (Figure 4A). We suppose that vitamin C and GO may interact and potentially alter GO toxicity. Interestingly, GO alone did not adversely affect female reproductive capacity; egg-laying and larval-hatching rates were comparable to the control group.

Despite the reduced number of eggs in the Vit. C groups, the high hatching success suggests that the eggs’ quality was unaffected, indicating a generally deteriorated condition of the females instead (Figure 4A). This improved hatching success could be linked to higher total Vg protein expression on the 15th day of adult life (Figure 2). While GO, Vit. C, or the GO20 + Vit. C mixture did not affect hatching success, reproductive toxicity is dose-dependent. Various studies have shown reproductive disorders at specific doses, such as poor spermatogenesis, higher apoptosis, and lower egg production [66,67,68]. Conversely, other studies showed no reproductive toxicity at different doses [69,70,71].

Dziewięcka et al. [72] reported that a high dose of GO (200 µg·mL^−1^) over ten days led to oxidative stress, DNA damage, apoptosis, and changes in the gut and testis of house crickets. Prolonged GO exposure at the same concentration significantly reduced reproductive capabilities [73]. In our project, using a ten times lower concentration of GO (20 µg·mL^−1^), no adverse effects on reproduction were observed.

The high hatching success suggests that the eggs were unaffected by GO or vitamin C. To assess egg quality, we analyzed the major energetic components (Figure 5). Proteins, the primary egg component, accounted for over 98%, with Vn constituting 70–90% of the total protein [74]. The eggs’ total protein, glucose, and glycogen content were similar across all groups (Figure 6). Differences were observed only in lipid content (Figure 7A). Vitamin C alone did not change egg composition, indicating its effect on *Vg* gene expression but not protein production or egg composition.

Significant changes in lipid content were noted on the 5th and 15th days in the GO20 + Vit. C and GO20 groups. Since lipids are the most energetic nutrients [75], increased lipid content may reflect a maternal investment in the developing embryo, potentially at the expense of the mother’s condition. This could explain the lower number of eggs laid but high hatching success in these groups (Figure 4). Higher lipid content equips the embryo for high-energy demands. The GO20 + Vit. C and GO20-fed groups had higher egg lipid content, possibly as a compensatory mechanism for the adverse effects of GO20 or GO20 + Vit. C.

## 4. Materials and Methods

### 4.1. Graphene Oxide Characteristics

An aqueous suspension of graphene oxide (GO) at 10 mg·mL^−1^ was purchased from Nanografi (Jena, Germany). This dispersion was diluted to 20 µg·mL^−1^ and sonicated for 1 min. The GO was then deposited on a silicon wafer for SEM and mica for AFM visualization, followed by overnight drying. The morphology was examined using a scanning electron microscope (SEM, Quanta FEG 250, FEI, Hillsboro, OR, USA) at 30 kV in high vacuum mode. Atomic force microscope (AFM) imaging was performed with an Agilent 5500 (Aligent Technologies, Santa Clara, CA, USA) in tapping mode. The zeta potential was measured using a Litesizer 500 (Anton Paar, Graz, Austria) at 25 °C, and the values were calculated using the Smoluchowski equation.

SEM and AFM methods visualized the GO flakes (Figure 7), showing mostly single-layer flakes with a thickness of 1.0 nm and an average area of about 2 µm^2^. The zeta potential of −28.8 mV confirmed the suspension’s stability. Detailed analysis of these GO samples, labeled as S3, is available in previous work by Dziewięcka et al. [76].

### 4.2. Characteristics of the Species

Acheta domesticus (Gryllidae, Orthoptera, Insecta) is a medium-sized insect known to be found all over the world. Since 2001, the University of Silesia in Katowice has been conducting selective breeding of house cricket [21,53,54,77]. This organism has many of the characteristics of a model organism, such as a short life cycle, high reproduction rate, and well-known physiology. It is easy to breed, and the breeding process is beneficial from an economic point of view [78]. All advantages of *A. domesticus* make it easy to use in research with sufficient repeats and convenient sample collection and processing.

### 4.3. Food Preparation: GO Food, Vitamin C Food, GO + Vitamin C Food, Control Food

Graphene oxide food (GO20) was prepared by dissolving GO in ultrapure water, mixing it with ground standard artificial food (Kanisan Q, Sano, Poland) to a concentration of 20 mg∙kg^−1^, then freezing at −70 °C and lyophilizing for 24 h. The food was stored in a dry, dark container.

Vitamin C food (Vit. C) was similarly prepared with L-ascorbic acid (Sigma-Aldrich, St. Louis, MI, USA) at 1 mg·g^−1^ of dry food, lyophilized, and stored at −20 °C. The food was changed every other day to avoid oxidation. Vitamin C concentrations were measured before (0.82 mg·g^−1^), mid-experiment (0.76 mg·g^−1^), and after (0.73 mg·g^−1^) according to the Polish standard protocol (PN-A-04019:1998) [79].

GO20 and vitamin C food (GO20 + Vit. C) combined GO (20 mg·kg^−1^) and ascorbic acid (1 mg·g^−1^) following the same preparation steps. The ascorbic acid concentration was measured before (0.77 mg·g^−1^), mid-experiment (0.70 mg·g^−1^), and after (0.59 mg·g^−1^).

Control food was prepared like GO20 food but without GO.

### 4.4. Experimental Model

The insects were kept in a laboratory breeding room under optimal conditions for house cricket growth and reproduction (temperature 28.8 ± 0.88 °C, photoperiod 12:12 L, humidity 20–45%). Four-day-old larvae were divided into four experimental groups: control, vitamin C (Vit. C), vitamin C with GO20 (GO20 + Vit. C), and GO20. They were provided with water and dedicated food ad libitum throughout their lives. On the 5th and 15th days of imago life, females were sacrificed, and samples (fat body, eggs) were collected for analysis with five repeats per group. On the 18th day of the imago stage, females and males were used to assess reproduction potential.

### 4.5. Sample Preparation and Measurement of Selected Parameters

#### 4.5.1. *Vg* Gene Relative Expression

Fat body samples were collected from female insects on the 5th and 15th days of adulthood, preserved in RNAlater (Sigma-Aldrich, St. Louis, MI, USA), and stored at −70 °C for future analysis. RNA extraction was performed using the GeneMATRIX Universal RNA Purification Kit (Eurx, Gdańsk, Poland), and RNA quality was assessed via standard gel electrophoresis. Subsequently, the NG dART RT-PCR kit (Eurx, Gdańsk, Poland) was utilized to synthesize the first strand of cDNA in a two-step RT-PCR reaction. The quality of cDNA was evaluated using NanoDrop 2000 (Waltham, MA, USA), and cDNA was stored at −20 °C for RT-PCR analysis.

Primers targeting the coding sequence fragment of the DUF1943 domain of Vitellogenin were designed using PrimerQuest software (Integrated DNA Technologies, Inc., Coralville, IA, USA, https://eu.idtdna.com/pages/tools/primerquest?returnurl=%2FPrimerQuest, accessed on 23 October 2023). The coding sequence of Vitellogenin was predicted from the genome of Acheta domesticus (GCA_031308135.1), provided by Dr. Brenda Oppert. Additionally, the beta-actin (KEGG Orthology identifier: K05692) coding sequence (Acc. GDVN01020720.1) was retrieved from the transcriptome assembly (GDVN00000000) [80] and identified using TransdecoderURL http://transdecoder.github.io (accesses on 10 October 2023) and KoalaGhost automatic annotation and KEGG mapping service [81]. The resulting putative protein sequence was found to be highly conserved across various insects, including Drosophila melanogaster [82]. RT-PCR was performed using the SG/ROX qPCR Master Mix (Eurx, Poland) with a temperature profile including holding stage at 95 °C for 5 s, cycling stage (35×) at 95 °C for 10 s, 59 °C for 15 s, and 72 °C for 30 s, holding stage at 60 °C for 60 s, and melt curve stage at 95 °C for 15 s, 60 °C for 60 s, and 95 °C for 30 s. The relative gene expression was calculated based on delta Ct.

#### 4.5.2. Total Vg Protein Expression

Fat body samples were collected in Eppendorf tubes containing PBS buffer with additives (100 μL, pH 7.4, 0.1 M; mercaptoethanol, protease inhibitor PMSF, NaN_3_). After homogenization and centrifugation (15,000× *g*, 10 min, 4 °C), the submitochondrial fraction was stored at −70 °C for future analysis.

An ELISA assay was employed to measure total Vg protein content, following a protocol similar to previous research [21,54]. Total protein in the fat body was assessed using the Bradford method [83]. Microtiter plates were coated with homogenates of identical protein concentrations and incubated overnight at 4 °C. The plates were washed with PBST and blocked with 1% BSA. After washing, anti-Vg antibody (produced in rabbit, GenScript, Piscataway, NJ, USA) was added to the plate (100 μL, 0.2 μg mL^−1^, 2 h, 37 °C). Following another washing step, a secondary antibody (Anti-Rabbit IgG Alkaline Phosphatase ALP, produced in goat, Sigma, Kanagawa, Japan) was added (100 μL, diluted 1000× in homogenizing buffer, 1 h, 37 °C). After washing, the pNPP substrate was added to detect alkaline phosphatase. The color reaction was measured using a microplate reader at 405 nm wavelength and expressed as mean optical density.

#### 4.5.3. Vg Protein Precursor and Subunits Expression

Western blot analysis for semi-quantitative Vg precursor and subunit expression followed protocols from previous works [21,54,84]. Samples from fat body tissue were prepared similarly to ELISA samples, with homogenates from five repeats per group. Protein content in the mixes was determined using the Bradford method [83]. Probes with identical protein concentrations were denatured and loaded onto gels along with a protein ladder marker. Pre-electrophoresis and exact electrophoresis were conducted, followed by electrotransfer to a nylon membrane. The membrane was blocked with 3% bovine serum albumin in Tris-buffered saline (TBS) and washed with TBS with Tween-20 (TBST). The primary antibody (anti-Vg produced in rabbit, GenScript) was added overnight at 4 °C, followed by washing and the addition of a secondary antibody (Rabbit IgG Alkaline Phosphatase produced in goat, Sigma) for 1 h at room temperature. After another washing step, the colorimetric substrate BCIP/NBT was used for the color reaction. The membrane was scanned and analyzed using ImageJ Software ( version: 1.54b), and the band area was measured.

#### 4.5.4. Egg Laying and Hatching Success: Eggs Laid per Female, Larvae Hatching Success per Female, Total Hatching Success

To assess reproduction success, three females and three males were placed in an insectary box, repeated five times per treatment group. Crickets could lay eggs for 48 h in boxes with wet soil, which were then transferred to the breeding room for approximately ten days. Larvae hatching success was determined using ImageJ Software every other day until hatching stopped. The remaining eggs were dried and counted for eggs laid per female. Egg-laying success was calculated as the sum of dried eggs in the soil and hatched larvae per female. Total hatching success was calculated as the percentage of hatched eggs to the total number of laid eggs in the experimental group.

#### 4.5.5. Major Energetic Components in Eggs: Lipids, Glucose, Glycogen, Total Protein

Cricket eggs’ major energetic components were measured at two points (the 5th day and the 15th day). The females (5 repeats per experimental group) were sacrificed, and the eggs were collected and placed in Eppendorf tubes and frozen at −70 °C for future analysis. The eggs’ lipids, glucose, glycogen, and protein content were measured, as by Foray et al. [83]. Briefly, the eggs were homogenized in lysis buffer and centrifuged (4 °C, 5 min, 180× *g*). Next, the homogenate was used to assess the protein content in the eggs. The rest of the sample, after centrifugation, was vortexed. A total of 180 µL of samples were placed in new Eppendorf tubes, and 20% sodium sulfate and a mixture of chloroform–methanol (1:2) were added. The suspension was centrifuged (4 °C, 15 min, 180× *g*). The homogenate was used to determine the content of glucose and lipids, and the pellet was used to measure glycogen content. The assessment of individual components was conducted using Bradford reagent (protein), lipids (vanillin reagent), glucose, and glycogen (anthrone reagent), and the procedures as in the publication of Foray et al. [83] were followed. Ready samples were read using a spectrophotometer UV–VIS (TECAN Infinite M200, Grödig, Austria) at the appropriate wavelength: proteins (595 nm), lipids (525 nm), glucose, and glycogen (625 nm). The blank sample in the performed determinations was a mixture of chloroform and methanol. The concentrations of the given substances were calculated by substituting the obtained absorbances into the equation from the previously prepared standard curves. Concentration values are shown in the unit mg·mL^−1^ of the homogenate.

### 4.6. Statistical Analysis

The results were analyzed using Statistica software (version: 13.3), including normality tests (Kolmogorov–Smirnov and Shapiro–Wilk) and the Levene test for homogeneity of variances. Parametric tests, such as ANOVA with post hoc LSD test (*p* < 0.05), were applied to investigate differences between groups.

This study tested several hypotheses:

**H1.0:** 
*Graphene oxide (GO) chronic intoxication does not alter the reproductive potential of A. domesticus at various molecular levels, including Vg gene and protein expression, egg laying and hatching success, and major egg energetic components. Antioxidant mechanisms may protect against or neutralize GO effects.*


**H1.1:** 
*GO intoxication affects A. domesticus reproductive potential, inducing oxidative stress and disrupting Vg gene and protein expression, egg laying and hatching success, and egg major components. Reproduction disorders may vary with insect age and antioxidant availability.*


**H2.0:** 
*Long-term ascorbic acid supplementation does not protect against GO effects, with no significant differences between GO-treated groups with or without vitamin C. The mechanism of GO toxicity may involve factors beyond oxidative stress, rendering vitamin C ineffective.*


**H2.1:** 
*Ascorbic acid supplementation may mitigate GO effects, leading to improved reproductive potential compared to GO-treated groups. Vitamin C reduces reactive oxygen species levels, preventing oxidative stress and maintaining reproduction parameters similar to controls.*


**H2.2:** 
*Ascorbic acid supplementation may have adverse effects on A. domesticus, disrupting Vg gene and protein expression and contributing to reproduction disorders.*


**H2.3:** 
*Ascorbic acid supplementation does not mitigate GO effects and may exacerbate adverse outcomes compared to GO treatment alone. Interaction between GO and vitamin C could lead to unpredictable effects on reproduction parameters.*


## 5. Conclusions

In the presented project, we focused on the possible remedial effect of vitamin C in invertebrate *Acheta domesticus*, which has been intoxicated with graphene oxide nanoparticles for their lives. We found no simple relationship between the Vitellogenin gene’s expression and the Vg protein content measured using ELISA and Western Blot techniques. Despite fewer eggs laid in the vitamin C groups, hatching success was high, and egg composition did not differ significantly between groups. The exceptions were the groups fed with GO20 and GO20 + Vit. C food, where there was a shift in the lipid content in eggs. We conclude that vitamin C did not affect egg quality. The results demonstrate that lower egg-laying success could have resulted from the female’s condition. Most likely, ascorbic acid impacts the level of *Vg* gene expression but does not affect the production of Vg protein or the quality of eggs laid. It can be assumed that the health effects of GO are likely dose-dependent. Low GO concentration in food used in this project did not cause adverse effects on investigated parameters. The relationship between GO toxicity and its concentration should be investigated more thoroughly.

## Figures and Tables

**Figure 1 molecules-29-04594-f001:**
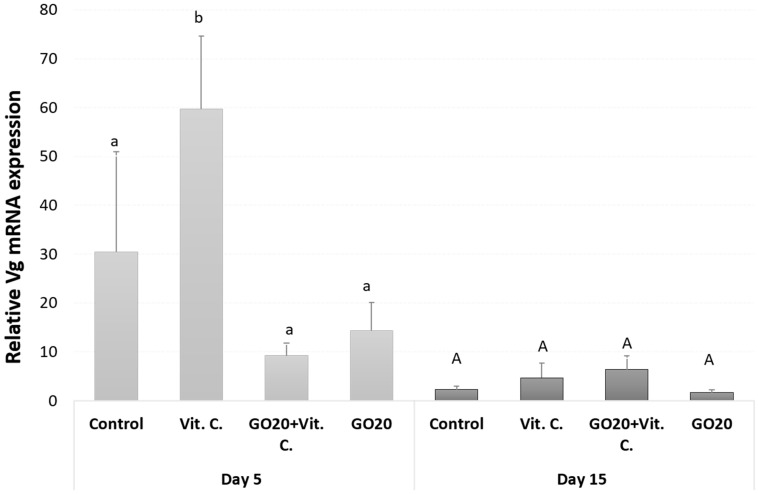
*Vg* gene expression levels in *A. domesticus* fat body on the 5th and 15th days of adult life. Data were shown as a relative expression compared to β-actin and expressed as means ± SE. Abbreviations: Control—animals fed uncontaminated food; Vit. C.—animals fed with Vitamin C in the food; GO20 + Vit. C.—animals fed with graphene oxide and Vitamin C; GO20—animals fed with graphene oxide; significant differences were measured using ANOVA (Fisher test; *p* < 0.05); different letters denote differences among the experimental groups within time points.

**Figure 2 molecules-29-04594-f002:**
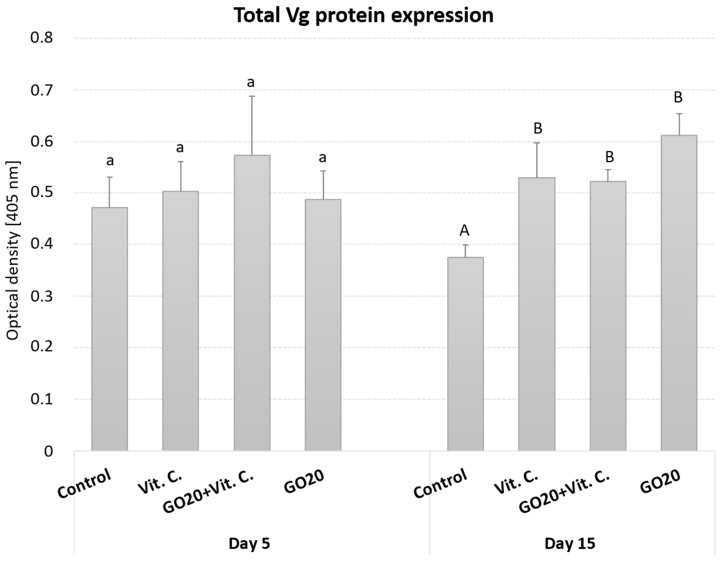
Total Vg protein content in the fat body of *A. domesticus* on the 5th and 15th days of adult life. Abbreviations: Control—animals fed uncontaminated food; Vit. C.—animals fed with Vitamin C in the food; GO20 + Vit. C.—animals fed with graphene oxide and Vitamin C in the food; GO20—animals fed with graphene oxide in the food; significant differences were measured using ANOVA (Fisher test; *p* < 0.05); different letters denote differences among the experimental groups and time points.

**Figure 3 molecules-29-04594-f003:**
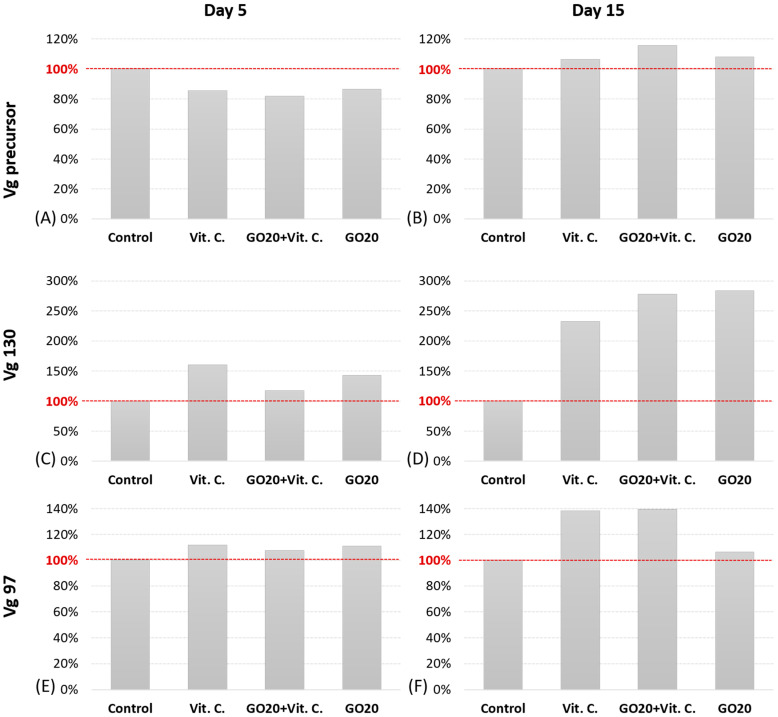
Semi-quantitative analysis of Vg protein in the fat body of *A. domesticus* on the 5th and 15th days of adult life. The graph presents the precursor’s Vg expression (~200 kDa) (**A**,**B**) and the subunits 130 kDa (**C**,**D**), 97 kDa (**E**,**F**). Abbreviations: Control—animals fed uncontaminated food; Vit. C.—animals fed with Vitamin C in the food; GO20 + Vit. C.—animals fed with graphene oxide and Vitamin C; GO20—animals fed with graphene oxide; Expression measured as band density compared to the reference (control, day 5th). All the groups were compared to controls presented as 100% (red line).

**Figure 4 molecules-29-04594-f004:**
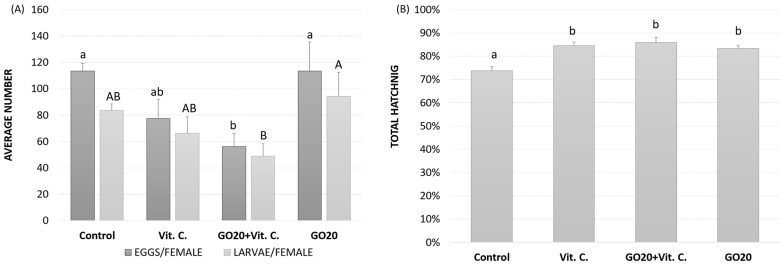
Egg laying (48 h) and hatching success of *A. domesticus*. (**A**) the average number of eggs laid per female (dark grey), the average number of larvae enclosed per female (light grey), and (**B**) the average total hatching success measured as a percent of enclosed eggs to the total number of laid eggs in the experimental group. Abbreviations: Control—animals fed uncontaminated food; Vit. C.—animals fed with Vitamin C in the food; GO20 + Vit. C.—animals fed with graphene oxide and Vitamin C; GO20—animals fed with graphene oxide; significant differences were measured using ANOVA (Fisher test; *p* < 0.05); different letters denote differences among the experimental groups (lower case: eggs/female, capital letters: larvae/female).

**Figure 5 molecules-29-04594-f005:**
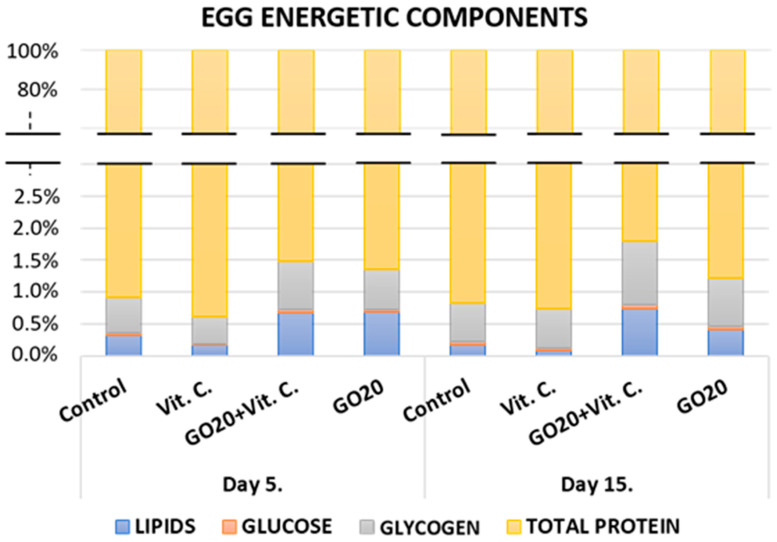
Share of major energetic components (lipids, glucose, glycogen, and total protein content) in the eggs of *A. domesticus* females collected on the 5th and 15th days of adult life. Abbreviations: Control—animals fed uncontaminated food; Vit. C.—animals fed with Vitamin C in the food; GO20 + Vit. C.—animals fed with graphene oxide and Vitamin C; GO20—animals fed with graphene oxide.

**Figure 6 molecules-29-04594-f006:**
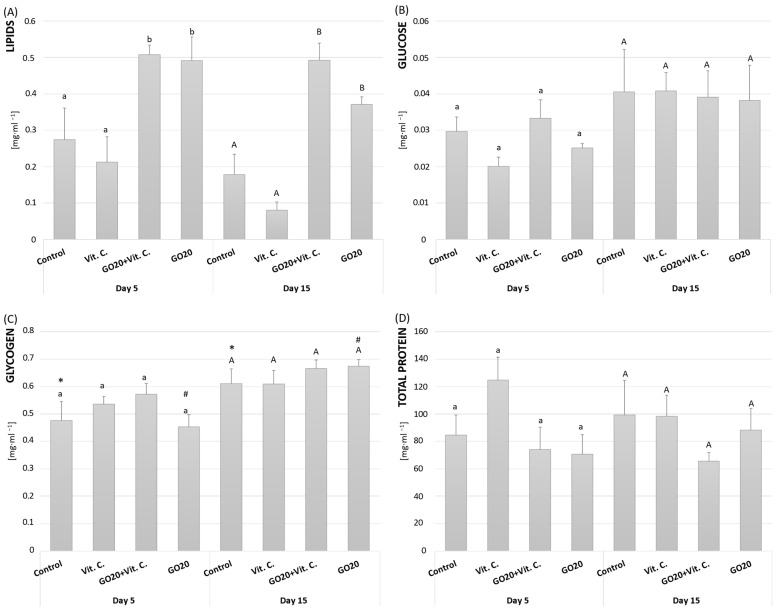
Major energetic components: lipids (**A**), glucose (**B**), glycogen (**C**), and total protein content (**D**) in the eggs of *A. domesticus* females collected on the 5th and 15th day of adult life. Abbreviations: Control—animals fed uncontaminated food; Vit. C.—animals fed with Vitamin C in the food; GO20 + Vit. C.—animals fed with graphene oxide and Vitamin C; GO20—animals fed with graphene oxide; significant differences were measured using ANOVA (Fisher test; *p* < 0.05); different letters denote differences among the experimental groups (small letters for the 5th day, and capital letters for the 15th day); an asterisk (*) and hashtag (#) show differences between corresponding groups on days 5th and 15th.

**Figure 7 molecules-29-04594-f007:**
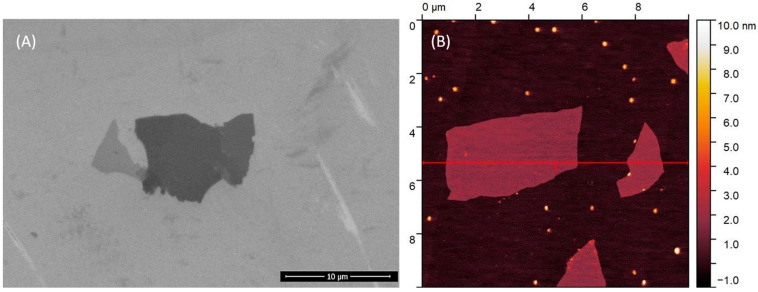
Image of graphene oxide (**A**) SEM Magnification: 10,000×; scale bar 10 µm; (**B**) AFM.

## Data Availability

Data are available on request.

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
