# Peer review of "Ascorbic Acid and Graphene Oxide Exposure in the Model Organism Acheta domesticus Can Change the Reproduction Potential"

_molecules, 2024, doi:10.3390/molecules29194594_

Round 1

Reviewer 1 Report

Comments and Suggestions for Authors

1. What is the concentration of Vit. C and GO20 used in the experiment? Why choose the concentration?

2. Why choose the 5 and 15 day?

3. The Acheta domesticus was not introduced clearly. And the scientific soundness is not clearly.

4. Low GO concentration in food did not cause adverse 20

effects, but its toxicity may depend on the concentration. The experiment data was not sufficient.

5. English should be improved further.

Comments on the Quality of English Language

Average.

Author Response

Dear Reviewer,

Thank you for our manuscript's insightful, constructive, and helpful review. Thank you for pointing out our mistakes and significant matters that we omitted while preparing our manuscript. Once again, from a new perspective, we looked at our work. We did our best to address the issue better. Below, we include detailed answers to your questions and explanations concerning your doubts.

Sincerely yours,

Barbara Flasz, Maria Augustyniak, and co-authors

Reviewer #1

1) What is the concentration of Vit. C and GO20 used in the experiment? Why choose the concentration?

Thank you for this question. The concentrations of graphene oxide and vitamin C are described in section ‘4.3. Food preparation: GO food, vitamin C food, GO+vitamin C food, control food’ in the materials and methods. We used a GO concentration of 20 mg∙kg-1 of dry food; in the case of vitamin C food, we used ascorbic acid in concentration 1 mg·g-1 of dry food. As vitamin C can oxidize, we decided on special conditions for its storage and supply. The ready-to-use food with vitamin C was stored at -20 degrees Celsius. The food in the insectaria was changed every two days to fresh food to avoid possible oxidation.

The concentrations we used are based on our previous experiences and a literature search. We have quite a lot of experience in testing the toxicity of nanomaterials, including graphene oxide. Our previous research examined different GO concentrations in various time frames. We were investigating the toxicity of high concentrations (200 mg∙kg-1) for a short time (7-10 days) and over a generation. In connection with the fact that in the natural environment, nanoparticles can occur, but instead in lower or trace concentrations, we also decided to conduct experiments with lower concentrations like 20 mg∙kg-1 of dry food or even lower (2 mg∙kg-1, 0.2 mg∙kg-1 of dry food) over generation and also during multiple generations (up to 6). Based on our work, these concentrations may be close to those in the environment.

In the case of vitamin C food, we also based on our experience and a reliable literature search to find vitamin C concentrations that would have a supplementary effect. We decided to prepare a pilot study with two vitamin C concentrations to assess the impact of vitamin C supplementation on insects (mortality, reproduction success, and cell health condition). We chose the 1 mg·g-1 of dry food concentration based on the results (the effect of vitamin C was observable and did not cause strong adverse effects). In our earlier papers, we wrote more extensively about the issues of GO and/or vitamin C concentration. Below are some references to some of our works that You might find interesting:

  • Flasz, B.; Dziewięcka, M.; Kędziorski, A.; Tarnawska, M.; Augustyniak, M. Multigenerational graphene oxide intoxication results in reproduction disorders at the molecular level of vitellogenin protein expression in Acheta domesticus. Chemosphere 2021, 280, doi:10.1016/j.chemosphere.2021.130772.
  • Flasz, B.; Dziewięcka, M.; Kędziorski, A.; Tarnawska, M.; Augustyniak, M. Vitellogenin expression, DNA damage, health status of cells and catalase activity in Acheta domesticus selected according to their longevity after graphene oxide treatment. Total Environ. 2020, 737, doi:10.1016/j.scitotenv.2020.140274.
  • Flasz, B.; Dziewięcka, M.; Kędziorski, A.; Tarnawska, M.; Augustyniak, J.; Augustyniak, M. Multigenerational selection towards longevity changes the protective role of vitamin C against graphene oxide-induced oxidative stress in house crickets. Pollut. 2021, 290, doi:10.1016/j.envpol.2021.117996.
  • Dziewięcka, M.; Witas, P.; Karpeta-Kaczmarek, J.; Kwaśniewska, J.; Flasz, B.; Balin, K.; Augustyniak, M. Reduced fecundity and cellular changes in Acheta domesticus after multigenerational exposure to graphene oxide nanoparticles in food. Total Environ. 2018, 635, 947–955, doi:10.1016/j.scitotenv.2018.04.207.
  • Dziewięcka, M.; Karpeta-Kaczmarek, J.; Augustyniak, M.; Rost-Roszkowska, M. Short-term in vivo exposure to graphene oxide can cause damage to the gut and testis. Hazard. Mater. 2017, 328, 80–89, doi:10.1016/j.jhazmat.2017.01.012.
  • Dziewięcka, M.; Karpeta-Kaczmarek, J.; Augustyniak, M.; Majchrzycki, Ł.; Augustyniak-Jabłokow, M.A. Evaluation of in vivo graphene oxide toxicity for Acheta domesticus in relation to nanomaterial purity and time passed from the exposure. Hazard. Mater. 2016, 305, 30–40, doi:10.1016/j.jhazmat.2015.11.021.
  • Dziewięcka, M.; Pawlyta, M.; Majchrzycki, Ł.; Balin, K.; Barteczko, S.; Czerkawska, M.; Augustyniak, M. The structure–properties–cytotoxicity interplay: A crucial pathway to determining graphene oxide biocompatibility. J. Mol. Sci. 2021, 22, doi:10.3390/ijms22105401.
  • Flasz, B.; Dziewięcka, M.; Ajay, A.K.; Tarnawska, M.; Babczyńska, A.; Kędziorski, A.; Napora-Rutkowski, Ł.; Ziętara, P.; Świerczek, E.; Augustyniak, M. Age-and Lifespan-Dependent Differences in GO Caused DNA Damage in Acheta domesticus. J. Mol. Sci 2022, 2023, 290, doi:10.3390/ijms24010290.

2) Why choose the 5 and 15 days?

Thank you for this comment. The time points were used based on our experimental work. The insects used in the project have been grown continuously in our laboratory for over 20 years. Over the years, we have been gradually collecting information about the biology of this species. Time points used in the experiment correspond to stages of insects' adult life. Day 5th of imago life stands for the stage when insects are adult, young, and ready to start reproduction processes. Day 15th of imago life is when adult insects can still reproduce, but they are starting to get old, and more energy is required to maintain homeostasis.

3) The Acheta domesticus was not introduced clearly. And the scientific soundness is not clearly.

Thank you for drawing attention to the research model. You are right. In the manuscript, the characteristics of the species (section 4.2. materials and methods) are described quite briefly. We fixed that mistake and hope the A. domesticus is introduced better and the importance of this species for scientific purposes is evident. That section in the manuscript was changed. At the same time, we did not want to expand this section excessively, as we have already written about A. domesticus as a model species in previous publications.

4) Low GO concentration in food did not cause adverse effects, but its toxicity may depend on the concentration. The experiment data was not sufficient.

You are right. Our statement was too confident, and we did not confirm it with sufficient results. We changed that in the manuscript.

The concentration of a nanoparticle may be significant in causing toxic effects. Although our results do not show this in a literal way, we can put forward a thesis that the toxicity of GO may depend on its concentration. A comparison of our previous results at higher concentrations clearly showed that GO has adverse effects, which can be seen in various cellular parameters such as viability, degree of apoptosis, level of oxidative stress, or even changes in the structure of cytoplasm and cellular organelles or DNA damage (references: above). However, lower concentrations such as the one used in this project only sometimes give obvious results of GO toxicity, which does not exempt us from the further necessity of testing this nanoparticle and recognizing it as safe. Many publications confirm our assumption about the dependence of GO concentration and toxicity. We allow ourselves to post references to some of them below:

  • Dou, T.; Chen, J.; Wang, R.; Pu, X.; Wu, H.; Zhao, Y. Complementary protective effects of autophagy and oxidative response against graphene oxide toxicity in Caenorhabditis elegans. Environ. Saf. 2022, 248, 114289, doi:https://doi.org/10.1016/j.ecoenv.2022.114289.
  • Zhang, X.; Wei, C.; Li, Y.; Li, Y.; Chen, G.; He, Y.; Yi, C.; Wang, C.; Yu, D. Dose-dependent cytotoxicity induced by pristine graphene oxide nanosheets for potential bone tissue regeneration. Biomed. Mater. Res. Part A 2020, 108, 614–624, doi:https://doi.org/10.1002/jbm.a.36841.
  • Dasmahapatra, A.K.; Powe, D.K.; Dasari, T.P.S.; Tchounwou, P.B. Assessment of reproductive and developmental effects of graphene oxide on Japanese medaka (Oryzias latipes). Chemosphere 2020, 259, 127221, doi:10.1016/J.CHEMOSPHERE.2020.127221.
  • Gurunathan, S.; Kang, M.-H.; Jeyaraj, M.; Kim, J.-H. Differential Cytotoxicity of Different Sizes of Graphene Oxide Nanoparticles in Leydig (TM3) and Sertoli (TM4) Cells., doi:10.3390/nano9020139.
  • Begum, P.; Ikhtiari, R.; Fugetsu, B. Graphene phytotoxicity in the seedling stage of cabbage, tomato, red spinach, and lettuce. Carbon N. Y. 2011, 49, 3907–3919, doi:https://doi.org/10.1016/j.carbon.2011.05.029.

5) English should be improved further.

Thank you for that comment. We have once again reviewed the text for English language accuracy. Additionally, MDPI provides language editing, which is routinely performed before the publication of an article. We trust that any errors we may have missed will be corrected at that stage.

Reviewer 2 Report

Comments and Suggestions for Authors

1.     The study mentions that Vitamin C may reduce GO, potentially altering its toxicity. Did the authors further validate the impact of this reduction on the experimental outcomes? If not, should this be addressed in the discussion section?

2.     The GO concentration used in the study is 20 µg/mL, but literature suggests that different concentrations of GO may exhibit varying levels of toxicity. Have the authors considered conducting experiments with different GO concentrations to provide a more comprehensive understanding of its effects on reproduction?

3.     The impact of Vitamin C on Vg gene expression and protein expression is inconsistent across different time points. Can the authors further explain this phenomenon and discuss the potential underlying biological mechanisms?

Comments on the Quality of English Language

1.     The abbreviation GO20+Vit.C should be consistent throughout the text. For instance, ensure that either Vit.C or Vitamin C is used uniformly across all instances.

2.     The capitalization of headings like Protein expression and Gene expression might be inconsistent. Ensure that the formatting of these headings is uniform throughout the document, such as Protein Expression and Gene Expression.

Author Response

Dear Reviewer,

Thank you for our manuscript's insightful, constructive, and helpful review. Thank you for pointing out our mistakes and significant matters that we omitted while preparing our manuscript. Once again, from a new perspective, we looked at our work. We did our best to address the issue better. Below, we include detailed answers to your questions and explanations concerning your doubts.

Sincerely yours,

Barbara Flasz, Maria Augustyniak, and co-authors

Reviewer #2

1) The study mentions that Vitamin C may reduce GO, potentially altering its toxicity. Did the authors further validate the impact of this reduction on the experimental outcomes? If not, should this be addressed in the discussion section?

Thank you for that valuable comment. You are right. This is a bold thesis we put forward. We did not validate the GO reduction in our experimental work. In the course of various experiments with GO in our laboratory, we observed that when GO suspensions and a vitamin C solution are combined (after a time, measured in hours to days), a gelatinous substance with a solid consistency form in the test tube, which aligns with the stages of graphene sponge formation. This process is related to the reduction of graphene oxide, where vitamin C is also used as a reducing agent. However, it is important to note that preparing cricket food did not involve the direct combination of GO suspension and vitamin C (both components were added separately to the ground feed). Therefore, the reduction of GO is much more complex, as it also involves the components of the feed matrix. Additionally, after consumption, new variables influence this process, such as changes in pH in the digestive tract or new compounds and molecules that appear from food digestion by digestive enzymes. Thus, proving our bold hypothesis at this research stage is very challenging.

We plan to focus on this topic by conducting semi-in vitro studies using a gut model we plan to design, which will allow us to control more parameters and variables. However, this is a more demanding study, requiring advanced equipment and the expertise of physicists and chemists. Nonetheless, we believe that in the future, we will be able to provide more information on the fate of GO and vitamin C in the gut. Because of these difficulties, we decided to remove this part of the discussion from the revised manuscript.

2) The GO concentration used in the study is 20 µg/mL, but the literature suggests that different concentrations of GO may exhibit varying levels of toxicity. Have the authors considered conducting experiments with different GO concentrations to provide a more comprehensive understanding of its effects on reproduction?

In our previous projects, we used different GO concentrations. We used high concentrations like 200 mg∙kg-1 or lower like 20 mg∙kg-1, 2 mg∙kg-1, and 0.2 mg∙kg-1 in different experimental variants. High concentrations caused toxicity, reproduction disorders, and even changes in the gonads. Lower GO concentrations used for whole life or even a few generations caused changes in reproduction success, for example, at the level of vitellogenin protein abundance. We did not conduct the experiment with vitamin C and different GO concentrations, mainly due to the limitations of breeding a large number of groups and synchronizing the sampling time. For technical reasons, this would be very difficult and even unfeasible. However, the results of such an extended experiment might be interesting. Thank you for that idea, which we will consider in the future. Below, you can find some of our previous publications with different graphene oxide concentrations in other variants of intoxication:

  • Flasz, B.; Dziewięcka, M.; Kędziorski, A.; Tarnawska, M.; Augustyniak, M. Multigenerational graphene oxide intoxication results in reproduction disorders at the molecular level of vitellogenin protein expression in Acheta domesticus. Chemosphere 2021, 280, doi:10.1016/j.chemosphere.2021.130772.
  • Flasz, B.; Dziewięcka, M.; Kędziorski, A.; Tarnawska, M.; Augustyniak, M. Vitellogenin expression, DNA damage, health status of cells and catalase activity in Acheta domesticus selected according to their longevity after graphene oxide treatment. Total Environ. 2020, 737, doi:10.1016/j.scitotenv.2020.140274.
  • Flasz, B.; Dziewięcka, M.; Kędziorski, A.; Tarnawska, M.; Augustyniak, J.; Augustyniak, M. Multigenerational selection towards longevity changes the protective role of vitamin C against graphene oxide-induced oxidative stress in house crickets. Pollut. 2021, 290, doi:10.1016/j.envpol.2021.117996.
  • Dziewięcka, M.; Witas, P.; Karpeta-Kaczmarek, J.; Kwaśniewska, J.; Flasz, B.; Balin, K.; Augustyniak, M. Reduced fecundity and cellular changes in Acheta domesticus after multigenerational exposure to graphene oxide nanoparticles in food. Total Environ. 2018, 635, 947–955, doi:10.1016/j.scitotenv.2018.04.207.
  • Dziewięcka, M.; Karpeta-Kaczmarek, J.; Augustyniak, M.; Rost-Roszkowska, M. Short-term in vivo exposure to graphene oxide can cause damage to the gut and testis. Hazard. Mater. 2017, 328, 80–89, doi:10.1016/j.jhazmat.2017.01.012.
  • Dziewięcka, M.; Karpeta-Kaczmarek, J.; Augustyniak, M.; Majchrzycki, Ł.; Augustyniak-Jabłokow, M.A. Evaluation of in vivo graphene oxide toxicity for Acheta domesticus in relation to nanomaterial purity and time passed from the exposure. Hazard. Mater. 2016, 305, 30–40, doi:10.1016/j.jhazmat.2015.11.021.
  • Dziewięcka, M.; Pawlyta, M.; Majchrzycki, Ł.; Balin, K.; Barteczko, S.; Czerkawska, M.; Augustyniak, M. The structure–properties–cytotoxicity interplay: A crucial pathway to determining graphene oxide biocompatibility. J. Mol. Sci. 2021, 22, doi:10.3390/ijms22105401.
  • Flasz, B.; Dziewięcka, M.; Ajay, A.K.; Tarnawska, M.; Babczyńska, A.; Kędziorski, A.; Napora-Rutkowski, Ł.; Ziętara, P.; Świerczek, E.; Augustyniak, M. Age-and Lifespan-Dependent Differences in GO Caused DNA Damage in Acheta domesticus. J. Mol. Sci 2022, 2023, 290, doi:10.3390/ijms24010290.

3) The impact of Vitamin C on Vg gene expression and protein expression is inconsistent across different time points. Can the authors further explain this phenomenon and discuss the potential underlying biological mechanisms?

Thank you for pointing out that exciting phenomenon. One would expect consistency and a clear relationship between gene expression and the amount of protein produced. Our results are different. In our opinion, the mechanism of vitamin C action may be related to the control of gene expression, including vitellogenin. Vitamin C is known for its antioxidant activity and is a cofactor in biological processes or immune responses. The other role of ascorbic acid is the modulation of gene expression. The most investigated ascorbic acid gene expression regulation is collagen synthesis. It is well documented that in the production of collagen type I, ascorbic acid increases the level of procollagen mRNA in two ways: by stimulating the transcription of the procollagen gene and by limiting the degradation of procollagen mRNA. Other genes are also stimulated by ascorbic acid molecules, showing the particular involvement of Vit. C in gene expression. Horovitz et al. (1989) showed that the acetylcholine receptor synthesis is stimulated by ascorbic acid. In other studies, the trend was that increasing dietary ascorbic acid caused a higher cytochrome P-450 content or changes in alkaline phosphatase levels. The described examples demonstrated that ascorbic acid may have a role at the transcriptional level. In the presented work, the Vg mRNA expression in insects during the 5th day of imago life was high. On the 15th day, the Vg gene was expressed higher than the control but with no statistical importance. In our opinion, lower expression of Vg mRNA during the 15th day may be related to the nature of Vg. Vitellogenin is sex, tissue, and stage-specific. On the 15th day of the imago stage, the Vg gene may not be any more intensively expressed due to the advanced age of the crickets.

The correlation between mRNA and protein abundance depends on biological and technical factors such as RNA secondary structure, regulatory proteins, regulatory sRNAs, ribosomal density, and ribosomal occupancy. Ascorbic acid used in this project seems to impact mRNA production significantly compared to protein production. Below, you can find some references:

  • Belin, S.; Kaya, F.; Burtey, S.; Fontes, M. Ascorbic Acid and Gene Expression: Another Example of Regulation of Gene Expression by Small Molecules?; Current genomics 2010; Vol. 11 (1), 52-57.
  • Horovitz, O.; Knaack, D.; Podleski, T.R.; Salpeter, M.M. Acetylcholine receptor alpha-subunit mRNA is increased by ascorbic acid in cloned L5 muscle cells: Northern blot analysis and in situ hybridization. J. Cell Biol. 1989, 108, 1823–1832, doi:10.1083/jcb.108.5.1823.
  • Horovitz, O.; Spitsberg, V.; Salpeter, M.M. Regulation of acetylcholine receptor synthesis at the level of translation in rat primary muscle cells. J. Cell Biol. 1989, 108, 1817–1822, doi:10.1083/jcb.108.5.1817.
  • Sullivan, T.A.; Uschmann, B.; Hough, R.; Leboy, P.S. Ascorbate modulation of chondrocyte gene expression is independent of its role in collagen secretion. J. Biol. Chem. 1994, 269, 22500–22506, doi:10.1016/S0021-9258(17)31675-7.
  • Torii, Y.; Hitomi, K.; Tsukagoshi, N. L-Ascorbic Acid 2-Phosphate Promotes Osteoblastic Differentiation of MC3T3-E1 Mediated by Accumulation of Type I Collagen. J. Nutr. Sci. Vitaminol. (Tokyo). 1994, 40, 229–238, doi:10.3177/jnsv.40.229.
  • Tufail, M.; Takeda, M. Molecular Characteristics of Insect Vitellogenins. J Insect Physiol 2008, 54, 1447–1458, doi:10.1016/j.jinsphys.2008.08.007.
  • Krumova, K.; Cosa G. Singlet Oxygen: Applications in Biosciences and Nanosciences. The Royal Society of Chemistry, 2016, ch. 1, pp. 1-21, doi: https://doi.org/10.1039/9781782622208
  • Suzuki, H.; Torii, Y.; Hitomi, K.; Tsukagoshi, N. Ascorbate-dependent elevation of mRNA levels for cytochrome P450s induced by polychlorinated biphenyls. Biochem. Pharmacol. 1993, 46, 186–189, doi:https://doi.org/10.1016/0006-2952(93)90365-4.

Comments on the Quality of English Language

4) The abbreviation “GO20+Vit.C” should be consistent throughout the text. For instance, ensure that either “Vit.C” or “Vitamin C” is used uniformly across all instances.

Thank you for that advice. We checked the manuscript and made corrections. We hope this will make the manuscript more understandable.

5) The capitalization of headings like “Protein expression” and “Gene expression” might be inconsistent. Ensure that the formatting of these headings is uniform throughout the document, such as “Protein Expression” and “Gene Expression.”

Thank you for that comment. We did the corrections.

Reviewer 3 Report

Comments and Suggestions for Authors

Dear Authors, 

I read carefully your manuscript, I appreciate it, and I made above some suggestions:

1. The period after the title should be removed. As well, all the punctuation marks (point) in the figures should be removed as well (Day 5 not Day 5.).

2. Line 163 - Total hatching percent (%) not "Total hatching success".

3. Line 467-469 - The methods regarding the lipids, glucose, glycogen, total protein quantification are not clearly described. Please improve the description, because is it not clear the protocol that you used.

4. The Conclusion section should be more specific. Please review it. I consider that expression like "Still, from our experience," should be changed in something like "Our results shows/demonstrates...". I mean that the conclusion section should be based on the results obtained in this study.

5. I have some concerns regarding the Reference sections. Please read it again carefully and pay attention that in some cited papers the Journal is missing or the writing stile is not uniform. Moreover, titles from 1984 or 1989, even if there are important in research development during the years, can be changed with some more recent study (if available).

Overall, I appreciate your work and I will agree with the publication of the manuscript after you proceed with the changes.

Best regards!

Comments on the Quality of English Language

Only some minor correction in the manuscript are necessary.

Author Response

Dear Reviewer,

Thank you for our manuscript's insightful, constructive, and helpful review. Thank you for pointing out our mistakes and significant matters that we omitted while preparing our manuscript. Once again, from a new perspective, we looked at our work. We did our best to address the issue better. Below, we include detailed answers to your questions and explanations concerning your doubts.

Sincerely yours,

Barbara Flasz, Maria Augustyniak, and co-authors

Reviewer #3

1) The period after the title should be removed. As well, all the punctuation marks (point) in the figures should be removed as well (Day 5 not Day 5.).

Thank you for that comment. The correction was made.

2) Line 163 - Total hatching percent (%) not "Total hatching success".

Thank you for that comment. The correction was made.

3) Line 467-469 - The methods regarding the lipids, glucose, glycogen, total protein quantification are not clearly described. Please improve the description, because is it not clear the protocol that you used.

Thank you for that comment. We rewrote the section, and we hope it is understandable now. Also, we have highlighted that the method used comes from a publication by Foray et al. (1985).

4) The Conclusion section should be more specific. Please review it. I consider that expression like "Still, from our experience," should be changed in something like "Our results shows/demonstrates...". I mean that the conclusion section should be based on the results obtained in this study.

Thank you for that valuable comment. The conclusions were rewritten.

5) I have some concerns regarding the Reference sections. Please read it again carefully and pay attention that in some cited papers the Journal is missing or the writing stile is not uniform. Moreover, titles from 1984 or 1989, even if there are important in research development during the years, can be changed with some more recent study (if available).

Thank you for your advice. We used dedicated software for citations. We read the references once more carefully and made the corrections. In some cases, we did not want to change the references for a few reasons: we used methods initially described in the publication; the publication is original research work where the phenomenon was described.

Overall, I appreciate your work and I will agree with the publication of the manuscript after you proceed with the changes.

Thank You very much for the insightful review.

Round 2

Reviewer 2 Report

Comments and Suggestions for Authors

No more question,the manuscript was good.